# FENDA-FL: Personalized Federated Learning on Heterogeneous Clinical Datasets

## Abstract

Federated learning (FL) is increasingly being recognized as a key approach to overcoming the data silos that so frequently obstruct the training and deployment of machine-learning models in clinical settings. This work contributes to a growing body of FL research specifically focused on clinical applications along three important directions. First, an extension of the FENDA method (Kim et al., 2016) to the FL setting is proposed. Experiments conducted on the FLamby benchmarks (du Terrail et al., 2022a) and GEMINI datasets (Verma et al., 2017) show that the approach is robust to heterogeneous clinical data and often outperforms existing global and personalized FL techniques. Further, the experimental results represent substantive improvements over the original FLamby benchmarks and expand such benchmarks to include evaluation of personalized FL methods. Finally, we advocate for a comprehensive checkpointing and evaluation framework for FL to better reflect practical settings and provide multiple baselines for comparison.

## 1 Introduction

It is well-established that the robustness and generalizability of machine-learning (ML) models typically grow with access to larger quantities of representative training data (Sun et al., 2017; Kaplan et al., 2020). While important in general settings, these properties are even more critical in clinical applications, where the consequences associated with performance degradation are heightened, as they directly impact patient care. For certain clinical tasks, studies have even produced estimates of minimum dataset sizes required to train models with sufficient utility to be useful in practical deployment (Hosseinzadeh et al., 2022). In practice, however, a large portion of health datasets are generated and held by multiple institutions. Sharing or centralizing the data is often discouraged, if not completely infeasible, due to strong regulations governing health data. Moreover, data annotation tends to be expensive, as it often requires experts, such as doctors. Thus, large, single-institution datasets are frequently unavailable.

Federated learning (FL), first introduced by McMahan et al. (2017), provides an avenue for training models in distributed data settings without requiring training data transfer. The focus of this work is so-called cross-silo FL (Kairouz et al., 2021), where participants represent a small number of reliable, large institutions with sufficient compute resources. In particular, we focus on the challenging setting of distributed heterogeneous datasets. Such datasets occur in many FL settings and are particularly common in clinical environments, as the statistical properties of local datasets are influenced by, for example, disparate patient populations and varying physical equipment.

Heterogeneous, or non-identically and independently distributed (non-IID), datasets often require special treatment in FL frameworks to avoid issues with convergence and to improve performance. This work proposes a novel extension of the frustratingly easy neural domain adaptation (FENDA) (Kim et al., 2016) method to the FL regime, termed FENDA-FL. This approach falls under the category of Personalized FL (Tan et al., 2022), where each client trains a model uniquely suited to its local distribution. To demonstrate the effectiveness of this approach, two clinically relevant dataset collections are considered. The first is a subset of tasks from the public FLamby benchmark (du Terrail et al., 2022a). The second set of experiments leverage two important tasks furnished by GEMINI (Verma et al., 2017), a large Canadian clinical data consortium.

The experiments to follow demonstrate that FENDA-FL often outperforms existing FL techniques, including a state-of-the-art personalized FL approach. Moreover, the experiments represent a signif-

icant extension of the FLamby benchmark along two dimensions. First, the results establish marked improvements in the performance of many FL approaches for the three tasks considered. Next, this work expands the benchmark results to include an evaluation of personal FL models. Finally, as part of the experimental design, we advocate for a more comprehensive evaluation framework for FL methods. This includes utilizing reasonable checkpointing strategies, bringing FL training into better alignment with standard ML practices and evaluating resulting model performance from several perspectives. Such a framework aims to improve the evaluation of new FL techniques and better reflect the setting of practical deployment.

## 2 RELATED WORK AND CONTRIBUTIONS

FL has been successfully applied in many contexts, including training models across mobile devices, wearable technologies, and financial data silos (Hard et al., 2018; Wang et al., 2019; Bogdanov et al., 2012). Moreover, it remains an active area of research along many dimensions, from communication efficiency (Passban et al., 2022; Isik et al., 2023) to improved optimization (Reddi et al., 2021). While the number of FL applications in clinical settings is growing (du Terrail et al., 2022b; Dayan et al., 2021), the approach is still relatively under-utilized, despite its potential benefits. This is due, in part, to a dearth of research and tooling focused on this important area. This paper aims to deepen fundamental understanding of how FL algorithms may be used to overcome clinical data silos and train ML models with potentially greater performance than even centralized data settings.[1]

A collection of work aimed at training personalized models through FL already exists (Tan et al., 2022; Fallah et al., 2020; Li et al., 2020b; Yu et al., 2020; Liang et al., 2020). These approaches aim to federally train models that produce elevated performance on each participating clients' individual data distribution. FENDA-FL falls under this category. We compare the performance of FENDA-FL against Adaptive Personalized Federated Learning (APFL) (Deng et al., 2020), a state-of-the-art personalized FL technique, showing that FENDA-FL performs comparably or better in all settings. In so doing, we extend the FLamby benchmark to include the evaluation of personalized FL approaches, which generally improve upon non-personal FL techniques, including those aimed at addressing non-IID training challenges. Some methods, (Li et al., 2021; Jeong & Hwang, 2022; Peterson et al., 2019), consider narrow settings or model architectures and are not considered.

FENDA-FL draws inspiration from Domain Adaptation (DA). Generally, DA involves learning a model from source datasets that generalizes to target datasets. During training, subsets of the target datasets, typically unlabeled or sparely labeled, are available. Federated DA considers the setting of distributed source and target datasets. Yao et al. (2022) consider a server-hosted source dataset and client-based, unlabelled datasets. In (Peng et al., 2019), the server holds the target dataset while labelled client datasets serve as the source. Finally, both source and target datasets may be distributed, as in (Shen et al., 2023; Song et al., 2020). Relatedly, Federated Domain Generalization focuses on the task of training a model on distributed, private source domains that generalizes to *unseen* target domains (Li et al., 2023). Such methods focus on training models for out-of-distribution clients rather than (personalized) models optimized for known client data. The settings above are related to those considered here but have different objectives and data settings. However, FENDA-FL establishes a useful conceptual bridge between FL and DA.

The study of FL in clinical settings has been expanding. Some works focus on the utility of FL for specific clinical applications (Andreux et al., 2020b;a; Chakravarty et al., 2021; Wang et al., 2023; Gunesli et al., 2021). The FLamby benchmark (du Terrail et al., 2022a) aims to standardize the underlying datasets in these studies to facilitate systematic evaluation of FL techniques in healthcare environments. In this work, we leverage several datasets from the FLamby benchmark to evaluate the proposed approach. In addition to establishing performance benchmarks for personalized FL methods, this work improves upon many of the originally reported baselines. Further, we extend the understanding of the practical use of FL in true clinical settings by measuring results on datasets from the GEMINI consortium, which are derived from up-to-date patient data at Canadian hospitals.

Finally, an important gap in the current literature surrounding FL is a set of guidelines for evaluating FL models that closely resemble practical deployment. This includes a clear discussion of checkpointing strategies and criteria required to demonstrate robust performance compared with baseline

---

[1]All experimental code is available at: URL withheld for double blind review.

approaches that do not leverage federated training. Many works (Li et al., 2020a; Karimireddy et al., 2020; McMahan et al., 2017; du Terrail et al., 2022a) simply perform a set number of server rounds and compare the resulting models along a small set of dimensions. In this paper, we suggest practical checkpointing approaches that more closely align FL training with standard ML model development, along with a comprehensive set of criteria for evaluating such models.

## 3 METHODOLOGY

As in (du Terrail et al., 2022a), the experiments to follow incorporate a number of existing FL techniques as baselines to which the proposed method is compared. In addition, as seen in Section 4, in many instances, the performance of these baselines often exceeds that observed in the original FLamby benchmarks, establishing new standards on the selected tasks. The methods considered are FedAvg (McMahan et al., 2017), FedAdam (Reddi et al., 2021), FedProx (Li et al., 2020a), SCAFFOLD (Karimireddy et al., 2020), and APFL (Deng et al., 2020).

The FedAvg algorithm, which applies weighted parameter averaging, remains a strong approach in many settings. FedAdam is a recent extension of FedAvg that incorporates server-side first- and second-order momentum estimates and has shown some robustness to data drift. In all experiments, $\beta_1 = 0.9$, $\beta_2 = 0.99$, and $\tau = 1\text{e-}9$. The FedProx and SCAFFOLD methods aim to address training issues associated with non-IID datasets by correcting local-weight drift through a regularization term or control variates, respectively. Each method trains a global set of weights, updated at each server round, to be shared by all clients. In the APFL approach, predictions are made through a convex combination of twin models. One is federally trained using FedAvg, while the other incorporates only local updates. The combination parameter, $\alpha$, is adapted throughout training using the gradients of the global and local models. This is a personalized FL approach where each client possesses a unique model. Two non-FL training setups are also considered. The first, central training, pools all training data to train a single model. The second is local training, where each client trains their own model solely on local data. This results in a single model per client with no global information.

For all experiments, clients use an AdamW optimizer (Loshchilov & Hutter, 2018) with default parameters for local training. The only exception is SCAFFOLD, which explicitly assumes an unmodified learning rate is applied and, thus, uses standard SGD. Hyper-parameters for each method, summarized in Table 1, are tuned. The ranges tested and best hyper-parameters for each method and dataset are detailed in Appendices A and B, along with the selection methodology.

| Method | FedAvg | FedAdam | FedProx | SCAFFOLD | APFL | FENDA-FL |
|---|---|---|---|---|---|---|
| Parameters | Client LR - | Client LR Server LR | Client LR $\mu$ | Client LR Server LR | Client LR $\alpha$ LR | Client LR - |

Table 1: A summary of hyper-parameters tuned for each approach, where LR denotes learning rate.

### 3.1 FENDA-FL

The proposed approach, FENDA-FL, is inspired by the FENDA method (Kim et al., 2016), originally designed for the setting of domain adaptation. Therein, each domain associated with a dataset has two feature extractors. One is exclusively trained on domain-specific data, while the other is shared and trained across all domains. The extracted features are then combined and processed by a domain-specific classification head. The central idea is that the shared feature extractor is free to learn to extract robust domain-agnostic features, while the domain-specific extractor learns features important in making accurate local predictions. Should the global features not prove useful for domain-specific prediction, the classification head can simply ignore those features and make predictions based solely on local features.

Adapting the FENDA setup to a federated setting is quite natural. Each client model consists of a global and local feature extraction module. The two sets of activations are combined and processed by a local classification network. Each client trains the full networks with local data during the client training phase. Thereafter, the clients only transmit the weights of the global feature extractors for server-side aggregation. In the experiments below, local and global features are fused by concate-

nation and server-side aggregation is performed through FedAvg. An illustration of FENDA-FL is shown in Figure 1. A technical discussion of FENDA-FL is found in Appendix J, including the forms of distribution drift handled by the architecture. FENDA-FL is more communication efficient than global methods with an equal number of weights, as only a subset of weights are exchanged.

In some ways, this setup resembles that of APFL, with important distinctions and advantages. Like APFL, FENDA-FL exchanges a subset of weights and predictions are produced through a combination of global and local sub-components. However, predictions in APFL are a convex combination of logits produced by duplicated models. In this respect, FENDA-FL is a more flexible approach - global and local modules need not have the same architectures. In clinical settings, this is useful for injecting inductive bias, as done in the Delrium task below, or incorporating unique local features. Moreover, the classification head in FENDA-FL is free to leverage non-linear transformations of the global and local features for prediction. Finally, APFL requires initialization of $\alpha$ and an associated learning rate. While APFL is fairly robust to such choices, they still have some impact on performance, as seen in Appendix G. The FENDA-FL approach also shares some properties with the Mixture of Experts method of (Li et al., 2020b). However, there a gating mechanism is used to learn a convex combination parameter, as used in APFL, rather than feature synthesis layers.

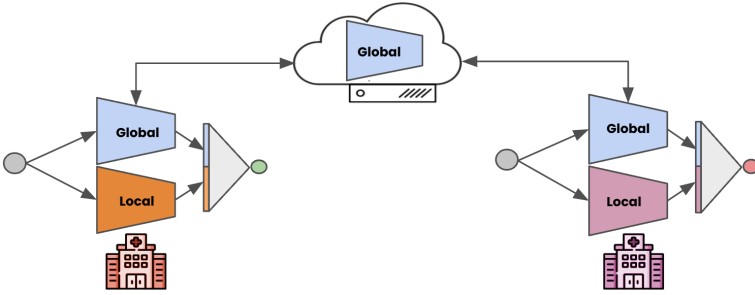

Figure 1: An illustration of FENDA-FL in a two client setting. Each client's model consists of local and global modules, followed by a local classifier. Global modules are exchanged with the server.

## 3.2 Clinical Datasets

In the experiments, five distinct clinical datasets are considered. Three are drawn from the FLamby benchmark datasets (du Terrail et al., 2022a) and two are privately held within the GEMINI consortium (Verma et al., 2017). Each represents a clinically relevant task incorporating data derived from real patients. The selected FLamby datasets are Fed-Heart-Disease, Fed-IXI, and Fed-ISIC2019. Fed-Heart-Disease is a binary classification task predicting whether a patient has heart disease from a collection of test measurements. Fed-IXI considers MR images of patients' heads with the target of 3D-segmentation of brain-tissue within the scans. Finally, Fed-ISIC2019 is a highly imbalanced, multi-class dermatology dataset consisting of 2D images. The task is to classify the images into one of eight different melanoma categories. More details on these datasets are found in Appendix A.

The two GEMINI datasets represent clinical data from patients discharged from seven hospitals in Canada. The data is anonymized by removing patient identifiers, yet still contains highly sensitive patient-level information. The costs and risks associated with centralizing such data are among the main motivations for this research. Given that the GEMINI consortium covers a huge set of variables associated with patient encounters, feature selection and preprocessing are conducted to prepare the source data for each task. Additional details are provided in Appendix B. The first task is in-hospital mortality prediction, which determines the probability of the patients' latest encounter ending in mortality based on admission records. The second task aims to predict the risk of delirium development in patients based on a wide variety of clinical measurements.

## 3.3 Federated Checkpointing and Evaluation

Presently, there is a gap in the FL literature thoroughly discussing practical considerations associated with model checkpointing strategies and robust evaluation. To this end, we propose a straightforward checkpointing strategy and evaluation framework for federally trained models, building on the

foundations in (du Terrail et al., 2022a) and better aligning with standard ML model practices. In many FL papers (du Terrail et al., 2022a; Li et al., 2020a; Karimireddy et al., 2020; Passban et al., 2022), federated training proceeds for a fixed number of server rounds. Thereafter, the resulting model is evaluated on held-out data, representing either a central or distributed test set. However, as with standard ML training, the model produced at the end of a training trajectory is not always optimal due to phenomena such as over-fitting.

We propose splitting each clients' dataset into train and validation sets. In the experiments, the split used is 80-20. The validation splits enable two distinct checkpointing approaches. After each round of local training and aggregation, the clients evaluate the model on their local validation sets. In *local checkpointing*, each client uses the local validation loss to determine whether a new checkpoint should be stored. Alternatively, in *global checkpointing*, the client losses are aggregated by the server using a weighted average. This average is used to checkpoint a global model to be shared by all clients. Note that, in both cases, the best model is not necessarily the one produced by the final server round. In the local case, clients are free to keep the model they deem best. In the global strategy, all clients share the same model from some point in the server rounds. Here, the personalized FL strategies do not have a single global model and only use local checkpointing.

To evaluate checkpointed models, we advocate for at least three separate baselines to be considered, two of which are part of the FLamby benchmarks. Models trained on *central*, or pooled, data are a common gold standard for FL performance. In the experiments, a single model is trained on data pooled from each of the clients and checkpointed based on a validation loss. The model is evaluated on each client's test set and the performance uniformly averaged. In *local* training, a client trains a model on its own training data. The model is shared with all clients and evaluated on their test data, including that of the original client. This setting mimics the scenario where, for example, a hospital distributes a model to other hospitals. Again, the measured metrics are averaged across clients. An important alternative setting, referred to here as *siloed*, is one where each client trains its own model and evaluates on its local test data. The results are then averaged across the clients. This simulates a strong baseline where, for example, the problem is important enough that each hospital has collected its own data to train a model. If an FL model does not produce better performance across clients, there is significantly less incentive to participate in collaborative training.

To quantify stability and smooth randomness in training, five FL training runs with distinct validation splits are performed for each method. Within each run, the performance across clients is averaged. The mean and 95% confidence intervals for this average are reported. When local checkpointing is used, each client loads its respective model to be evaluated on that client's test data.

## 4    RESULTS

In this section, all experiments use the checkpointing and evaluation strategies detailed in Section 3.3. Trainable parameter counts for the models are documented in Appendix C. All APFL training initializes $\alpha = 0.5$. For the FLamby experiments, 15 server rounds are performed for Fed-Heart-Disease and Fed-ISIC2019, while 10 are performed for Fed-IXI. Each client completes 100 local training steps within each round. Training settings for centralized and local training for each dataset are discussed in Appendix A. The loss functions, batch sizes, models, and metrics are all chosen to match the FLamby benchmark settings, unless otherwise specified. For both GEMINI tasks, 50 server rounds are performed. The clients in Mortality and Delirium prediction complete 2 and 4 local epochs, respectively, at each server round. Training setups are discussed in Appendix B.

### 4.1    FLAMBY: FED-HEART-DISEASE

The baseline model for Fed-Heart-Disease is logistic regression. The feature space for this dataset is only 13 dimensions. Hence, this model incorporates only 14 parameters. For APFL, a two layer dense neural net (DNN) is used for both the global and local model with a ReLU in between. For FENDA-FL, the feature extractors are single linear layers with a ReLU activation. The classification head is a linear layer followed by a sigmoid. These networks have 151 and 152 total parameters, respectively. Experiments using two-layer DNNs with an equivalent number of parameters for the other strategies did not improve performance, as seen in Appendix D. The results, displayed in Table 2, show that FENDA-FL is the best performing approach, followed by APFL. Not only

does FENDA-FL significantly outperform all other FL strategies, but it also outperforms centralized training by a large margin. Furthermore, FENDA-FL and APFL are the only FL mechanisms to beat siloed training, where each client trains and evaluates on its own data. The results also suggest that local checkpointing has some advantages for this task, except for SCAFFOLD. In Appendix E, ablation studies are carried out demonstrating the advantage of using federated checkpointing over a standard fixed server-round strategy.

## 4.2 FLAMBY: FED-IXI

As a 3D segmentation task, the metric for Fed-IXI is a DICE score. The model architecture is a 3D U-Net (Cicek et al., 2016) and the implementation in FLamby is used. The model produces a set of features for each voxel in the 3D image in the form of channels at the beginning and end of the U-Net. These channels are fused through a skip connection and concatenation. The per-voxel channel contents are then used to produce a binary classification for each voxel. For a fair comparison of other approaches to FENDA-FL and APFL, which require architecture modifications, the number of such channels in the U-Net is increased to 12, compared with the FLamby default of 8. The models used for FENDA-FL and APFL each produce 8 channels. The FENDA-FL architecture uses the U-Net without a classifier for both the global and local modules. The activations are concatenated and fed into a classification layer. As seen in Table 2, each strategy produces a model that performs this task well. However, there is a clear advantage to centralized training and an even larger one from FL. Models trained with FedAvg and FedProx surpass centralized training. Moreover, both personalized FL approaches produce models that surpass all other FL strategies and centralized training, with APFL beating FENDA-FL by a small margin. Additional experiments in Appendix F show that this gap can be closed with thoughtful architecture design. As with Fed-Heart-Disease, a small advantage is seen in using local checkpointing in most methods.

| | Fed-Heart-Disease Mean Accuracy | | Fed-IXI Mean DICE | | Fed-ISIC2019 Mean Balanced Accuracy | |
|---|---|---|---|---|---|---|
| | Server | Local | Server | Local | Server | Local |
| FedAvg | 0.724 (0.000) | 0.724 (0.000) | 0.9845* (0.0001) | 0.9846* (0.0001) | 0.657 (0.004) | 0.644 (0.004) |
| FedAdam | 0.719 (0.000) | 0.742 (0.000) | 0.9811* (0.0002) | 0.9811* (0.0002) | 0.647 (0.013) | 0.637 (0.012) |
| FedProx | 0.716 (0.000) | 0.721 (0.000) | 0.9842* (0.0001) | 0.9844* (0.0001) | 0.633 (0.022) | 0.652 (0.022) |
| SCAFFOLD | 0.711 (0.000) | 0.682 (0.000) | 0.9827* (0.0000) | 0.9827* (0.0000) | **0.672*** (0.013) | 0.669 (0.010) |
| APFL | | 0.801* (0.006) | | **0.9864*** (0.0002) | | 0.608 (0.011) |
| FENDA-FL | | **0.815*** (0.000) | | 0.9856* (0.0001) | | 0.607 (0.011) |
| Silo | | 0.748 (0.000) | | 0.9737 (0.0040) | | 0.634 (0.025) |
| Central | | 0.732 (0.000) | | 0.9832 (0.0002) | | 0.683 (0.020) |
| Client 0 | | 0.735 (0.000) | | 0.9648 (0.0123) | | 0.549 (0.014) |
| Client 1 | | 0.649 (0.000) | | 0.9722 (0.0004) | | 0.387 (0.029) |
| Client 2 | | 0.589 (0.000) | | 0.9550 (0.0015) | | 0.554 (0.010) |
| Client 3 | | 0.629 (0.000) | | - | | 0.444 (0.016) |
| Client 4 | | - | | - | | 0.326 (0.018) |
| Client 5 | | - | | - | | 0.285 (0.039) |

Table 2: Performance metrics for the FLamby tasks. Values in parentheses are 95% confidence-interval (CI) radii. Server and local refer to the checkpointing strategies and Silo metrics are measured as described in Section 3.3. Bold denotes the best performing federated strategy, while underlined values are second best. Asterisks imply a value is better than siloed training considering CIs. Note that each task has a different number of clients. For example, Fed-Heart-Disease has only four.

These results stand in contrast to those reported in the original FLamby benchmark. There, all FL strategies severely under-performed even local training. In that work, a standard SGD optimizer is used for client-side optimization, whereas we use AdamW. It should also be noted that the models for this task, along with those of Fed-ISIC2019, employ Batch Normalization layers. When using FedAdam, it is critical that stats tracking be turned off for these layers to avoid issues during the aggregation process. This phenomenon is discussed in Appendix H. We speculate that a related issue may have impeded the convergence of the FedOpt family in their experiments.

## 4.3 FLAMBY: FED-ISIC2019

The model used for Fed-ISIC2019 is EfficientNet-B0 (Tan & Le, 2019), which has been pretrained on ImageNet, with a linear final layer. For APFL, the full model is used for the global and local

components. For FENDA-FL, the final linear layer is removed and the remaining architecture is used for the local and global feature extractors. The features are then passed to a two-layer DNN with a ReLU activation in between the layers. For a fair comparison, the bottom 13 layers of EfficientNet-B0 are frozen for APFL and FENDA-FL.

The results in Table 2 show that all FL strategies perform fairly well. SCAFFOLD provides the best results. While it does not surpass centralized training, it is quite close. Further, each of the non-personalized FL methods equals or surpasses siloed evaluation. Again, these results improve upon those of the original FLamby benchmark, where none of the FL methods showed strong performance. This led the authors to conclude that federated training was not a useful approach for this task. The results presented here compellingly support the opposite conclusion.

**Mean Balanced Accuracy: Fed-ISIC2019**

|  | Client 0 | Client 1 | Client 2 | Client 3 | Client 4 | Client 5 |
|---|---|---|---|---|---|---|
| Client 0 | 0.67781 | 0.58954 | 0.59666 | 0.38725 | 0.40478 | 0.63506 |
| Client 1 | 0.22467 | 0.69782 | 0.38833 | 0.20257 | 0.37306 | 0.43795 |
| Client 2 | 0.37171 | 0.75305 | 0.6937 | 0.41446 | 0.47196 | 0.62035 |
| Client 3 | 0.2715 | 0.46201 | 0.50126 | 0.63792 | 0.44094 | 0.34767 |
| Client 4 | 0.16894 | 0.27198 | 0.21094 | 0.21286 | 0.66951 | 0.41896 |
| Client 5 | 0.20175 | 0.3235 | 0.26333 | 0.22589 | 0.27184 | 0.42481 |
| Central | 0.67198 | 0.70762 | 0.7391 | 0.60563 | 0.62952 | 0.74261 |
| SCAFFOLD | 0.59532 | 0.76551 | 0.78579 | 0.63607 | 0.48994 | 0.75803 |
| FENDA-FL | 0.60359 | 0.65925 | 0.73191 | 0.62158 | 0.63057 | 0.39748 |
| APFL | 0.60023 | 0.65677 | 0.72411 | 0.63113 | 0.62107 | 0.41355 |

Source Domain (rows) / Target Domain (columns)

Figure 2: Generalization measurements for a selection of FL and non-FL approaches. Rows correspond to a model trained using either client-only data, centralized data, or with the indicated FL method. Columns denote the performance of that model on the local test data of the named client.

For this task, both personalized FL approaches under-perform. While the performance is much better than any of the local models, they are the only FL methods that fail to reach siloed accuracy. Some insight into where these approaches fall short is found in Figure 2. SCAFFOLD outperforms FENDA-FL and APFL on test data drawn from Client 1 and Client 2. This is reversed when considering Client 4. However, the most significant difference is that both FENDA-FL and APFL perform poorly on test data from Client 5. Notably, Client 5 has the smallest dataset by a large margin; see Table 6 in Appendix A. Moreover, the data appears to be of poor quality, with locally trained models only achieving a balanced accuracy of $0.425$ on the test set. Alternatively, models trained only on data from Client 0 generalize well to Client 5. We hypothesize that the low quality of Client 5's data impedes personalized FL model training. In future work, we aim to consider ways of overcoming this issue. Some potential avenues include constructing additional loss terms to ensure the global and local feature extractors remain individually useful for classification or using a global FL approach to initialize the FENDA-FL model components as a warm start for further fine-tuning. Interestingly, for this task, local checkpointing appears disadvantageous compared to global checkpointing.

## 4.4 GEMINI: MORTALITY

For mortality prediction, the goal is to determine whether patients will experience a mortality event during their hospital stay. A collection of electronic-health-record and administrative features such as interventions, imaging descriptions, and blood transfusion data, are used to train a DNN. A standard binary cross-entropy loss is used for training. Except for the personalized methods, all approaches, including locally and centrally trained models, use the same DNN architecture with five layers. APFL and FENDA-FL require different model structures, but the number of trainable parameters remains the same. FENDA-FL's global and local modules each have two layers with half

the number of neurons of the full-size model. The classifier module has three layers. In APFL, half of the full model is used for each of the global and local components.

The performance of mortality models is assessed based on two metrics: accuracy and AUC/ROC. The results are shown in Table 3. FL methods improve upon locally trained models, reducing the performance gap to the centrally trained model. This is especially true when considering local model performance measured via AUC/ROC. In general, there is little difference between local and global checkpointing for this task, except for FedAvg, where global checkpointing improves performance.

In terms of accuracy, FedAvg is the only standard FL method that rivals the personalized FL algorithms and approaches siloed evaluation. On the other hand, most methods surpass the siloed AUC/ROC evaluation metric, nearing centrally trained model performance. Specifically, FedProx and SCAFFOLD beat siloed training based on AUC/ROC, but did not in terms of accuracy. This may be due to the class imbalance in this dataset. In both measures, FENDA-FL outperforms APFL, though the advantage in terms of accuracy is not statistically significant, and is the only approach to surpass both siloed evaluation metrics. As a result, we observe that FENDA-FL is robust to whether the data distribution is in favor of the local or shared models.

| | GEMINI Mortality | | | |
| | Mean Accuracy | | Mean AUC/ROC | |
| | Server | Local | Server | Local |
|---|---|---|---|---|
| FedAvg | 0.8976 (0.0002) | 0.8971 (0.0006) | 0.8180* (0.0005) | 0.8177* (0.0005) |
| FedAdam | 0.8954 (0.0000) | 0.8954 (0.0000) | 0.8128 (0.0005) | 0.8123 (0.0008) |
| FedProx | 0.8957 (0.0004) | 0.8956 (0.0003) | 0.8174* (0.0004) | 0.8172* (0.0006) |
| SCAFFOLD | 0.8955 (0.0003) | 0.8954 (0.0000) | **0.8186*** (0.0002) | 0.8184* (0.0002) |
| APFL | | 0.8978 (0.0004) | | 0.8138 (0.0017) |
| FENDA-FL | | **0.8980** (0.0004) | | 0.8181* (0.0001) |
| Silo | | 0.8978 (0.0009) | | 0.8140 (0.0012) |
| Central | | 0.8981 (0.0005) | | 0.8193 (0.0005) |
| Client 0 | | 0.8966 (0.0014) | | 0.8052 (0.0030) |
| Client 1 | | 0.8930 (0.0025) | | 0.7657 (0.0064) |
| Client 2 | | 0.8943 (0.0060) | | 0.7591 (0.0116) |
| Client 3 | | 0.8954 (0.0000) | | 0.7994 (0.0034) |
| Client 4 | | 0.8955 (0.0005) | | 0.7974 (0.0037) |
| Client 5 | | 0.8973 (0.0014) | | 0.7878 (0.0043) |
| Client 6 | | 0.8954 (0.0000) | | 0.8019 (0.0035) |

Table 3: Performance metrics for GEMINI Mortality. Values in parentheses are 95% confidence-interval (CI) radii. Server and local refer to the checkpointing strategies and Silo metrics are measured as described in Section 3.3. Bold denotes the best performing federated strategy, while underlined values are second best. Asterisks imply a value is better than siloed training considering CIs.

## 4.5 GEMINI: DELIRIUM

The goal of this task is to determine whether delirium occurred during a hospital stay in order to facilitate quality measurement and benchmarking, because delirium is a leading cause of preventable harm in hospitals. Labeling delirium cases is labor-intensive, requiring specialized medical knowledge. Even in the GEMINI consortium, only a relatively small population has been labeled, see Table 8. Features such as lab measurements, medications, and diagnosis are used to perform binary classification. More details are available in Appendix B.2.

Compared to the Mortality task, a larger DNN with six layers and more neurons is used for all non-personalized FL methods. APFL's global and local components are each half the size of the full-sized model. FENDA-FL's model consists of a two-layer and a four-layer feature extractor as the local and global modules, respectively. The classifier is a linear transform followed by a sigmoid activation that maps global and local module outputs to the final output for each client. The number of trainable parameters is kept approximately constant across all methods, as documented in Appendix C. Note that the FENDA-FL network is asymmetric. Appendix F discusses experimental results associated with various architecture choices for this task and Fed-IXI.

Results are shown in Table 4, including both mean accuracy and AUC/ROC. Non-personalized approaches such as FedProx and FedAvg outperform siloed evaluation in both metrics, while SCAF-FOLD and FedAdam under-perform, despite an extensive hyper-parameter sweep. Both personal-

ized methods also outperform the siloed evaluation metric, with FENDA-FL improving upon APFL. Across all methods, based on mean accuracy, only FedProx's globally checkpointed model surpasses the performance of FENDA-FL. Alternatively, based on mean AUC/ROC, FENDA-FL is the best approach for this use case, followed by FedProx. Interestingly, two locally trained models, Client 2 and Client 5, beat siloed evaluation in AUC/ROC. Table 8 shows that these clients have the largest number of samples, which could explain why their models appear to generalize so well.

| | GEMINI Delirium | | | |
| | Mean Accuracy | | Mean AUC/ROC | |
| | Server | Local | Server | Local |
|---|---|---|---|---|
| FedAvg | 0.7987 (0.0127) | 0.8025 (0.0130) | 0.8302* (0.0146) | 0.8274* (0.0078) |
| FedAdam | 0.7689 (0.0036) | 0.7688 (0.0127) | 0.7897 (0.0078) | 0.7881 (0.0055) |
| FedProx | **0.8095*** (0.0036) | 0.8056 (0.0037) | 0.8488* (0.0043) | 0.8504* (0.0042) |
| SCAFFOLD | 0.7480 (0.0000) | 0.7480 (0.0000) | 0.5491 (0.0469) | 0.5491 (0.0469) |
| APFL | | 0.8031 (0.0052) | | 0.8430* (0.0062) |
| FENDA-FL | | 0.8064 (0.0036) | | **0.8518*** (0.0017) |
| Silo | | 0.7936 (0.0102) | | 0.8037 (0.0072) |
| Central | | 0.8114 (0.0068) | | 0.8458 (0.0026) |
| Client 0 | | 0.7653 (0.0070) | | 0.7977 (0.0128) |
| Client 1 | | 0.7781 (0.0063) | | 0.7795 (0.0161) |
| Client 2 | | 0.7805 (0.0036) | | 0.8185* (0.0021) |
| Client 3 | | 0.7634 (0.0062) | | 0.7539 (0.0055) |
| Client 4 | | 0.7924 (0.0061) | | 0.8179 (0.0076) |
| Client 5 | | 0.8028 (0.0040) | | 0.8429* (0.0018) |

Table 4: Performance metrics for GEMINI Delirium. Values in parentheses are 95% confidence-interval (CI) radii. Server and local refer to the checkpointing strategies and Silo metrics are measured as described in Section 3.3. Bold denotes the best performing federated strategy, while underlined values are second best. Asterisks imply a value is better than siloed training considering CIs.

As an additional test of non-IID robustness in the studied FL methods, we add two local data processing steps to each clients' data pipeline, which significantly increases the heterogeneity of the local data distributions. The results of this experiment, discussed in detail in Appendix I, highlight the robustness and effectiveness of personalized methods, especially FENDA-FL, in the context of extremely heterogeneous data. There, FENDA-FL is the only method to recover siloed evaluation.

## 5 DISCUSSION AND CONCLUSIONS

In this work, we make substantive contributions to FL research with a specific focus on clinical applications. An extension of the FENDA framework to the FL setting is proposed and experiments demonstrate its effectiveness for clinical tasks and robustness to heterogeneous distributions. The experimental results also represent an extension of the FLamby benchmark along two dimensions for the selected tasks. The experiments show that FL methods provide convincing utility for the Fed-IXI and Fed-ISIC2019 tasks, which was not the case in the original benchmark experiments. Furthermore, the reported results provide strong personalized FL baselines not present in the original benchmarks. Finally, we outline a comprehensive checkpointing and evaluation framework, building on the original FLamby benchmark.

In future work, we aim to continue to improve the performance of FENDA-FL through, for example, the application of warm-start strategies and auxiliary losses. Another interesting avenue is incorporating so-called Generalization Adjustment, proposed in (Zhang et al., 2023), into the FL methods to improve performance. Further, we intend to investigate incorporating differential privacy, specifically in the context of clinical applications, and consider methods to improve the inherent utility-privacy trade-offs, which are even more critical in healthcare settings. Finally, we plan to extend the experimental results to include additional clinical tasks and larger pools of FL participants to better quantify the effects of participant scaling.

## 6 REPRODUCIBILITY

In this work, we have endeavoured to provide significant detail around the experiments and reported results to ensure that reproduction is as straightforward as possible. All code associated with the

FLamby experiments, including the scripts to perform hyper-parameter sweeps, hyper-parameter selection, evaluation, and visualizations is publicly released with this manuscript. All relevant parameters used for the experiments are extensively discussed in Sections 3 and 4, along with Appendices A and B. All artifacts generated by the FLamby experiments, including logs and checkpoints, are available upon request. These are not publicly available solely due to the size of the artifacts. For the GEMINI experiments, as much detail as possible is provided around the datasets, models, and hyper-parameters to ensure that practitioners with access to these private datasets can easily reproduce the experiments. Details around preprocessing and feature engineering are provided in Appendix B. We are happy to share code with individuals who have access to GEMINI. We welcome reproducibility studies and extensions of this work. However, due to the sensitive nature of the data and GEMINI's closed environment, the experimental code is not publicly released.

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

## A FLAMBY EXPERIMENTAL DETAILS AND HYPER-PARAMETERS

In this section, details are provided around each of the selected datasets in the FLamby benchmark. For additional details on items such as preprocessing and feature engineering, see (du Terrail et al., 2022a). We use the library provided with the FLamby benchmark to replicate the training settings, loss functions, model architectures, and other experimental components for consistency. Table 5 details the hyper-parameter sweeps and best values used for the results reported in Section 4, which differ from those of the original FLamby paper. The best hyper-parameters are chosen by computing the lowest aggregated validation loss seen by the server during FL training, averaged over five separate training runs. The aggregated validation loss is computed by averaging the loss on each client's validation set, weighted by the number of training examples held by each client. For central and local training, a setup identical to the FLamby benchmark is used. As a result, the best performing parameters optimized in the benchmark are applied for each task.

In selecting the FL algorithms for the FLamby and GEMINI experiments, FedAdam is chosen as representative of the FedOpt family (Reddi et al., 2021) because it performed well in the original

| Method | Task Parameters | Fed-Heart-Disease Range | Best | Fed-IXI Range | Best | Fed-ISIC2019 Range | Best |
|---|---|---|---|---|---|---|---|
| FedAvg | Client LR | 0.1 to 1e-5 | 0.1 | 0.1 to 1e-5 | 1e-3 | 0.1 to 1e-5 | 1e-3 |
| FedAdam | Client LR | 0.1 to 1e-5 | 1e-5 | 0.1 to 1e-5 | 1e-3 | 0.1 to 1e-5 | 1e-5 |
|  | Server LR | 0.1 to 1e-5 | 0.1 | 1.0 to 1e-4 | 0.01 | 1.0 to 1e-4 | 1e-3 |
| FedProx | Client LR | 1e-3 to 1e-4 | 1e-3 | 0.1 to 1e-5 | 1e-3 | 0.1 to 1e-5 | 1e-3 |
|  | $\mu$ | 1.0 to 0.01 | 0.01 | 1.0 to 1e-3 | 1e-3 | 1.0 to 1e-3 | 0.1 |
| SCAFFOLD | Client LR | 0.1 to 1e-5 | 0.1 | 0.1 to 1e-5 | 0.1 | 0.1 to 1e-5 | 0.01 |
|  | Server LR | 0.1 to 1e-5 | 0.1 | 1.0 to 1e-4 | 1.0 | 1.0 to 1e-4 | 1.0 |
| APFL | Client LR | 0.1 to 1e-5 | 0.1 | 0.1 to 1e-5 | 1e-3 | 0.1 to 1e-5 | 1e-4 |
|  | $\alpha$ LR | 1.0 to 1e-4 | 0.1 | 1.0 to 1e-4 | 1.0 | 1.0 to 1e-4 | 0.01 |
| FENDA-FL | Client LR | 0.1 to 1e-5 | 1e-3 | 0.1 to 1e-5 | 1e-3 | 0.1 to 1e-5 | 1e-3 |

Table 5: The hyper-parameters tuned for each FL method and FLamby dataset pair. For the "Range" column, the sets progress in orders of magnitude. For example, the range 0.1 to 1e-4 is the set of parameters {0.1, 0.01, 1e-3 1e-4}. The "Best" hyper-parameter values are chosen based on the average aggregated validation loss during federated training over five different training runs.

FLamby experiments. Similarly, the Cyclic method is not considered here as it performed poorly across all original FLamby benchmarks.

### A.1 Fed-Heart-Disease

The Fed-Heart-Disease dataset consists of four clients. The loss function is binary cross-entropy and the metric is standard accuracy. A batch size of 4 is used for all training. Central and local training uses a learning rate of 0.001 over 50 epochs with an AdamW optimizer. The client dataset distributions are shown in Table 6.

### A.2 Fed-IXI

The Fed-IXI dataset incorporates data from three distinct centres. Two different scanning systems produced the 3D images in the dataset. The brain scans are labeled with binary segmentation masks. For training, a DICE loss (Dice, 1945) is used with an $\epsilon = 10^{-9}$. The metric is the DICE score between the predicted and labeled segmentations. A constant batch size of 2 is used for all training. Both central and local training apply a learning rate of 0.001 and an AdamW optimizer for 10 epochs. Dataset statistics for each of the clients are shown in Table 6.

### A.3 Fed-ISIC2019

The Fed-ISIC2019 dataset is split across six different clients. Three of the client datasets are derived from the same source, the Medical University of Vienna, but were generated by different imaging devices. A weighted focal loss function (Lin et al., 2020) is used to fine-tune the EfficientNet-B0 architectures and balanced accuracy is measured to assess performance. All training runs use a batch size of 64. Central and local training uses an AdamW optimizer with a learning rate of 5e-4 over 20 epochs. As seen in Table 6, the number of data-points held by each client varies dramatically.

## B GEMINI Experimental Details and Hyper-parameters

GEMINI is a research network in which hospitals share data to a central repository. As the repository contains detailed patient data, access to GEMINI data is granted only after approval, in accordance with the network's data sharing agreement and research ethics protocols.[2] Similar to the FLamby experiments, the best hyper-parameters are chosen based on the minimum average validation loss

---

[2]GEMINI data access may be requested at `https://www.geminimedicine.ca/access-data`

| Dataset | Number | Client | Train Size | Test Size |
|---|---|---|---|---|
| Fed-Heart-Disease | 0 | Cleveland's Hospital | 199 | 104 |
| | 1 | Hungarian Hospital | 172 | 89 |
| | 2 | Switzerland Hospital | 30 | 16 |
| | 3 | Long Beach Hospital | 85 | 45 |
| Fed-IXI | 0 | Guy's Hospital | 262 | 66 |
| | 1 | Hammersmith Hospital | 142 | 36 |
| | 2 | Institute of Psychiatry | 54 | 14 |
| Fed-ISIC2019 | 0 | Hospital Clínic de Barcelona | 9930 | 2483 |
| | 1 | ViDIR Group, Medical University of Vienna (MoleMax HD) | 3163 | 791 |
| | 2 | ViDIR Group, Medical University of Vienna (DermLite FOTO) | 2691 | 672 |
| | 3 | Skin Cancer Practice of Cliff Rosendahl | 1807 | 452 |
| | 4 | Memorial Sloan Kettering Cancer Center | 655 | 164 |
| | 5 | ViDIR Group, Medical University of Vienna (Heine Dermaphot) | 351 | 88 |

Table 6: Train and test set sizes across clients in each of the FLamby benchmark datasets, reproduced from (du Terrail et al., 2022a). Each source institution represents one client in the FL process.

computed by the server. This average loss is the weighted average of clients' model's loss in each round of FL training. The sweep space and best hyper-parameters found for each of the FL methods are reported in Table 7. Hyper-parameter sweeps are also performed for central and local model training for both GEMINI tasks, the results of which are also found in the table.

While both the Mortality and Delirium prediction tasks are drawn from GEMINI consortium data, the data pre-processing steps result in two distinct task-specific datasets. Table 8 displays the client data distribution for both tasks. An overview of the data preparation steps is provided in the following sections. However, access to the code is restricted in accordance with privacy regulations.

### B.1 MORTALITY PREDICTION

For this task, there are seven participating hospitals, each representing a client with its own private data. For all experiments, the batch size is set to 64. The number of samples for mortality prediction is large compared to other tasks. A careful analysis of the available features resulted in 35 features selected for this task, followed by a standard data processing pipeline, which includes data vectorization and scaling. A non-exhaustive list of selected features includes: readmission status, whether the patient is transferred from an acute care institution, length of stay in days, CD-10-CA classification code, counts of previous encounters, diagnosis type, and age. Categorical features require special preprocessing in a federated setting. For clients to consistently map such features to a numerical space, we assume that the server has oracle knowledge of all the possible feature values and provides one-hot mappings to all clients.

### B.2 DELIRIUM PREDICTION

For this task, data is drawn from six hospitals. Each hospital participates in FL as a single client. As seen in Table 8, the number of samples in this task is smaller than in mortality prediction. This is because delirium labels are provided through a rigorous and time-intensive process of medical record review by a trained expert (Inouye et al., 2005). This is one of the only validated methods

| Method | Task Parameters | Mortality Range | Best | Delirium Range | Best |
|---|---|---|---|---|---|
| FedAvg | Client LR | 0.01 to 1e-4 | 1e-3 | 0.01 to 1e-4 | 1e-3 |
| FedAdam | Client LR | 0.01 to 1e-4 | 1e-4 | 0.01 to 1e-4 | 1e-4 |
|  | Server LR | 1.0 to 1e-3 | 1e-3 | 1.0 to 1e-3 | 1e-3 |
| FedProx | Client LR | 1.0 to 1e-4 | 0.01 | 0.1 to 1e-4 | 1e-3 |
|  | $\mu$ | 1.0 to 0.01 | 0.1 | 1.0 to 0.01 | 0.1 |
| SCAFFOLD | Client LR | 0.01 to 1e-4 | 0.01 | 0.01 to 1e-4 | 1e-3 |
|  | Server LR | 1.0 to 1e-3 | 1.0 | 1.0 to 1e-4 | 1e-3 |
| APFL | Client LR | 0.1 to 1e-4 | 0.01 | 0.01 to 1e-5 | 1e-4 |
|  | $\alpha$ LR | 1.0 to 1e-3 | 0.01 | 1.0 to 1e-3 | 0.01 |
| FENDA-FL | Client LR | 0.01 to 1e-5 | 1e-4 | 0.01 to 1e-5 | 1e-4 |
| Local 0 | LR | 0.01 to 1e-4 | 0.01 | 0.01 to 1e-4 | 0.01 |
| Local 1 | LR | 0.01 to 1e-4 | 1e-3 | 0.01 to 1e-4 | 0.01 |
| Local 2 | LR | 0.01 to 1e-4 | 0.01 | 0.01 to 1e-4 | 1e-4 |
| Local 3 | LR | 0.01 to 1e-4 | 0.01 | 0.01 to 1e-4 | 1e-3 |
| Local 4 | LR | 0.01 to 1e-4 | 0.01 | 0.01 to 1e-4 | 1e-3 |
| Local 5 | LR | 0.01 to 1e-4 | 0.01 | 0.01 to 1e-4 | 1e-3 |
| Local 6 | LR | 0.01 to 1e-4 | 0.01 | — | — |
| Central training | LR | 0.01 to 1e-5 | 1e-4 | 0.01 to 1e-5 | 1e-4 |

Table 7: The hyper-parameters tuned for each FL method, along with local and central model training, across GEMINI tasks. For the "Range" column, the sets progress in orders of magnitude. For example, the range 0.01 to 1e-4 is the set of parameters {0.01, 1e-3 1e-4}. The "Best" hyper-parameter values are chosen based on the average aggregated validation loss during federated training over five different training runs.

| Dataset | Number | Train Size | Test Size | Dataset | Number | Train Size | Test Size |
|---|---|---|---|---|---|---|---|
|  | 0 | 13234 | 3308 |  | 0 | 660 | 101 |
|  | 1 | 21018 | 5254 |  | 1 | 590 | 99 |
| GEMINI Mortality | 2 | 14180 | 3545 | GEMINI Delirium | 2 | 1218 | 199 |
|  | 3 | 15251 | 3813 |  | 3 | 494 | 76 |
|  | 4 | 14441 | 3610 |  | 4 | 574 | 109 |
|  | 5 | 12718 | 3180 |  | 5 | 1392 | 258 |
|  | 6 | 23820 | 5955 |  |  |  |  |

Table 8: Train and test set sizes across clients in the GEMINI datasets. Source institution names are withheld for privacy. Each institution represents one client in the FL process.

for such labeling, as it is poorly captured in routine administrative or clinical data. A wide variety of medical features is used, such as count of room transfer and ICU transfer, whether the patient has entered an intensive care unit since their admission, length of stay in days, laboratory-based Acute Physiology Score, ICU length of stay in hours, duration of the patient's stay in the ER in hours, whether admission was via ambulance, triage level, total number of transfusions, type of blood product or component transfused, medication names, text description of radiology images, among many others. The feature engineering pipeline results in $1,119$ features.

Concretely, the data pre-processing steps are the following. One-hot encoding is used for categorical features such as gender, discharge disposition, and admission category. A TF-IDF vectorizer is used to vectorize text-based columns such as radiology image and medication descriptions. A `MaxAbsScaler` is used to scale the data. Finally, during training, resampling is performed to balance negative and positive samples and increase the dataset size. Resampling is done separately in each client based on the client's data. Note that resampling is not performed on the test datasets. Similar to the mortality prediction task, we also assume oracle knowledge in the server, which enables clients to map categorical features into the same space. If clients individually process such features, there is a high chance of misalignment, resulting in heterogeneous feature spaces. Finally, during training, the batch size is 64 for all methods.

## C    TRAINABLE MODEL PARAMETERS IN EXPERIMENTS

Table 9 provides a comparison of the trainable parameters present in the models used in the FLamby and GEMINI experiments. The "standard" column refers to models trained in all settings, excluding APFL and FENDA-FL. For a fair comparison, we have sought to keep the number of trainable parameters across different architectures close to ensure that, for example, APFL and FENDA-FL models do not have out-sized expressiveness.

| Dataset | Standard | APFL | FENDA-FL |
|---|---|---|---|
| Fed-Heart-Disease | 14 or 152 | 152 | 151 |
| Fed-IXI | $1,106,520$ | $984,624$ | $984,620$ |
| Fed-ISIC2019 | $4,017,796$ | $4,631,088$ | $4,775,016$ |
| Mortality | $222,784$ | $222,784$ | $222,784$ |
| Delirium | $8,988,736$ | $8,988,736$ | $8,717,448$ |

Table 9: Trainable parameter counts for the model architectures used in the FLamby and GEMINI experiments. Standard refers to models used in all settings outside of APFL and FENDA-FL.

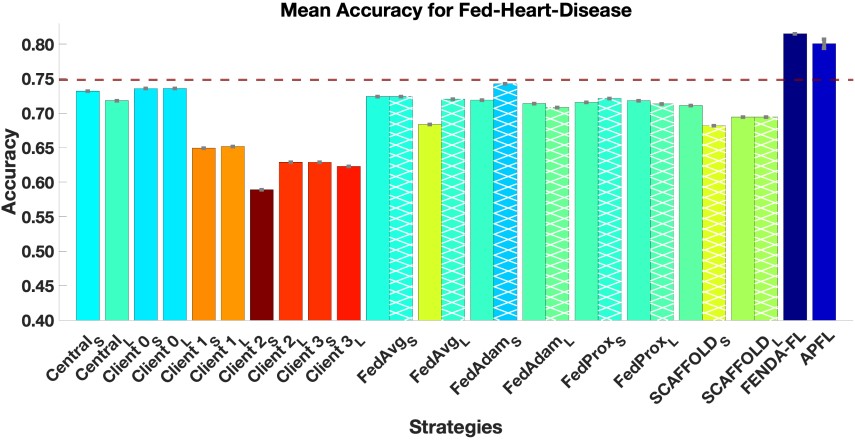

Figure 3: Mean accuracy for the Fed-Heart-Disease task. This figure shows results for small and large models with 14 and 152 parameters, respectively, for strategies other than FENDA-FL and APFL. Strategies sub-scripted with an S use small models and those with an L use the large models.

## D    FED HEART DISEASE: LARGE MODEL PERFORMANCE

Figure 3 displays the performance of both small and large models for various training strategies on Fed-Heart-Disease. The small models are logistic regression with only 14 trainable parameters. The large versions are two-layer DNNs with 152 parameters. The models used by FENDA-FL and APFL

are DNN-based with 151 and 152 parameters, respectively. For strategies other than FENDA-FL and APFL, the large model variants generally do not improve performance. Note that validation-based checkpointing, as described in Section 3.3, is applied in all cases. Therefore, it is unlikely that this phenomenon is due to over-fitting.

# E    FEDERATED CHECKPOINTING ABLATION EXPERIMENTS

To quantify the benefits of the federated checkpointing strategies proposed in Section 3.3, we compare the results against the widely used fixed-rounds strategy, where only the model from the final FL round is saved. For non-personalized approaches, this is the final aggregated model on the server. For personalized approaches, the final-round client-side model is saved after any server-side aggregation. These checkpoints are referred to as "Latest" in Table 10.

For Fed-Heart-Disease, there is a clear advantage to federated checkpointing, especially the local version, for all strategies except of SCAFFOLD. Notably, APFL and FENDA-FL benefit significantly from local federated checkpointing. For Fed-IXI, two experiments are conducted. In the first, the optimal hyper-parameters for each strategy are applied during training. The benefits of local checkpointing are less clear in this setting, though APFL and FENDA-FL do exhibit small improvements. In examining the training trajectories, many of the aggregated validation losses do not show signs of overfitting during the FL rounds, likely reducing the advantage of federated checkpointing.

| Fed-Heart-Disease Mean Accuracy | | | |
|---|---|---|---|
| | Latest | Server | Local |
| FedAvg | 0.706 (0.000) | **0.724** (0.000) | **0.724** (0.000) |
| FedAdam | 0.711 (0.000) | 0.719 (0.000) | **0.742** (0.000) |
| FedProx | 0.703 (0.000) | 0.716 (0.000) | **0.721** (0.000) |
| SCAFFOLD | **0.717** (0.000) | 0.711 (0.000) | 0.682 (0.000) |
| APFL | 0.777 (0.013) | | **0.801** (0.006) |
| FENDA-FL | 0.802 (0.010) | | **0.815** (0.000) |

| Fed-IXI Optimal Params. Mean DICE | | | |
|---|---|---|---|
| | Latest | Server | Local |
| FedAvg | 0.9845 (0.0002) | 0.9845 (0.0001) | **0.9846** (0.0001) |
| FedAdam | **0.9815** (0.0001) | 0.9811 (0.0002) | 0.9811 (0.0002) |
| FedProx | 0.9841 (0.0001) | 0.9842 (0.0001) | **0.9844** (0.0001) |
| SCAFFOLD | **0.9828** (0.0000) | 0.9827 (0.0000) | 0.9827 (0.0000) |
| APFL | 0.9858 (0.0000) | | **0.9864** (0.0002) |
| FENDA-FL | 0.9848 (0.0026) | | **0.9856** (0.0001) |

| Fed-IXI Sub-Optimal Params. Mean DICE | | | |
|---|---|---|---|
| | Latest | Server | Local |
| FedAvg | 0.9766 (0.0006) | **0.9788** (0.0006) | **0.9788** (0.0006) |
| FedAdam | 0.9202 (0.0091) | 0.9483 (0.0056) | **0.9488** (0.0054) |
| FedProx | 0.9295 (0.0356) | 0.9548 (0.0131) | **0.9558** (0.0136) |
| SCAFFOLD | **0.4360** (0.0000) | **0.4360** (0.0000) | **0.4360** (0.0000) |
| APFL | 0.9780 (0.0007) | | **0.9793** (0.0005) |
| FENDA-FL | 0.9705 (0.0075) | | **0.9790** (0.0004) |

Table 10: Checkpoint ablation results comparing federated checkpointing strategies with saving models after a fixed number of FL rounds. Values in parentheses are 95% confidence-interval radii.

To consider the setting where the optimal number of rounds is not well calibrated, we consider performance when training with sub-optimal hyper-parameters for each strategy. For all strategies,

a client-side learning rate of $0.1$ is applied. For FedAdam and SCAFFOLD, a server-side learning rate of $0.1$ and $1.0$, respectively, is used. FedProx sets $\mu = $ 1e-3, and APFL applies a learning rate for $\alpha$ of $1.0$. In this context, a significant advantage is seen when using federated checkpointing. Again, local federated checkpointing outperforms the server-side analogue. It is also interesting to note that federated checkpointing results in a measurable reduction in variance for all strategies.

## F    FENDA-FL ARCHITECTURE STUDIES

As discussed in Section 3.1, an important advantage of the FENDA-FL setup is that the local and global feature extraction modules need not be symmetric. Moreover, the latent spaces may also differ in size. This allows for the injection of inductive bias through architecture choice. For example, in situations where global models perform well, one may assign more parameters and a larger latent space to FENDA-FL's global feature extractor. In this section, we show that such choices can produce optimal performance.

| Layers (G, L) | Latent (G, L) | Parameters | Acc. | AUC/ROC |
|:---:|:---:|:---:|:---:|:---:|
| $4, 4$ | $64, 64$ | $8, 631, 424$ | $0.7926$ $(0.0071)$ | $0.8222$ $(0.0036)$ |
| $4, 2$ | $64, 64$ | $8, 729, 600$ | $0.8024$ $(0.0037)$ | $0.8368$ $(0.0039)$ |
| $4, 2$ | $128, 8$ | $8, 717, 448$ | $\mathbf{0.8064}$ $(0.0036)$ | $\mathbf{0.8518}$ $(0.0017)$ |
| $2, 4$ | $64, 64$ | $8, 729, 600$ | $0.7935$ $(0.0073)$ | $0.8153$ $(0.0059)$ |
| $2, 4$ | $8, 128$ | $8, 717, 448$ | $0.7935$ $(0.0073)$ | $0.8153$ $(0.0059)$ |

Table 11: Results corresponding to different FENDA-FL architecture variations on the Delirium task. "Layers (G, L)" and "Latent (G, L)" refer to the number of dense layers assigned to the global and local feature extractors, respectively, along with the size of the corresponding latent spaces.

Results for variations in the FENDA-FL model architecture for the Delirium task are reported in Table 11. Aside from FENDA-FL, the best performing strategy for this task is FedProx with an accuracy of $0.8095$ and AUC/ROC of $0.8504$, using a model with $8, 988, 736$ parameters. From the table, it is evident that assigning more layers and a larger latent space to the global FENDA-FL module is beneficial. Given that a global model is performing this task well, such an emphasis makes sense. However, the use of an, albeit smaller, personal module allows FENDA-FL to approach FedProx in terms of accuracy and surpass it in terms of AUC/ROC, despite having fewer parameters. Note that, in the imbalanced layer setting, the number of parameters lost, through reducing the depth of one module, are approximately transferred to the deeper module by increasing the layer sizes.

| Strategy | Depth (G, L) | Latent (G, L) | Parameters | DICE |
|:---:|:---:|:---:|:---:|:---:|
| APFL | $3, 3$ | $11, 11$ | $1, 859, 928$ | $0.9855$ $(0.0001)$ |
| | $3, 3$ | $8, 8$ | $984, 624$ | $0.9864$ $(0.0002)$ |
| FENDA-FL | $3, 3$ | $11, 11$ | $1, 859, 924$ | $0.9859$ $(0.0001)$ |
| | $3, 2$ | $14, 14$ | $1, 824, 740$ | $0.9864$ $(0.0000)$ |
| | $2, 3$ | $14, 14$ | $1, 824, 740$ | $\mathbf{0.9865}$ $(0.0000)$ |
| | $3, 2$ | $16, 8$ | $2, 070, 692$ | $0.9859$ $(0.0000)$ |
| | $2, 3$ | $8, 16$ | $2, 070, 692$ | $0.9860$ $(0.0000)$ |

Table 12: Results corresponding to different FENDA-FL architecture variations on the Fed-IXI task. "Depth (G, L)" and "Latent (G, L)" refer to the number of projection layers used in the global and local U-net feature extractors, respectively, along with the size of the latent space for each voxel.

Table 12 shows results for FENDA-FL architecture perturbations for the Fed-IXI task. While the performance changes are fairly small, there is a slight advantage to using a larger local feature extraction module for this task. Doing so brings the FENDA-FL model results in line with the

best performing strategy, APFL, for this problem. Note that APFL does not appear to benefit from additional parameters in this setting.

## G  APFL LEARNING RATE STUDIES

The stability of the APFL algorithm with respect to the choice of the learning rate for $\alpha$ across each of the FLamby datasets is considered in this section. Figure 4 shows the average metric across five APFL training runs when the learning rate for $\alpha$ is varied. In all cases, $\alpha$ is initially set to $0.5$. The performance of models trained with APFL is fairly robust to variations in the learning rate across each dataset, especially Fed-IXI. However, an impact on accuracy and balanced accuracy for the Fed-Heart-Disease and Fed-ISIC2019 tasks is observed. In both settings, a full percentage point of accuracy is lost for poorly chosen learning rates compared to the optimal learning rate.

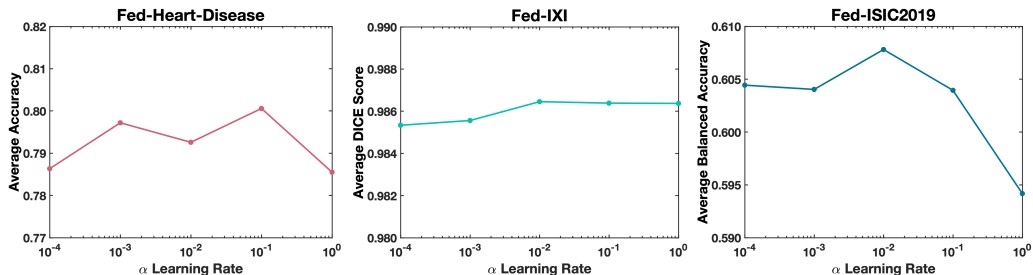

Figure 4: Changes in the average metric attained by models trained with APFL when the learning rate for $\alpha$ is varied. The method is fairly robust to such changes, but there is still a measurable impact on the performance when such learning rates are poorly chosen.

---

**Algorithm 1:** Simplified FedAdam calculation with a single parameter and constant updates, $\tilde{x} = 0.1$, from participating clients over 30 server rounds.

---

$x_t = x_0, \, m_t = 0, \, v_t = 0$
**for** $t$ *from 1 to 30* **do**
$\quad \Delta_t = \tilde{x} - x_t$
$\quad m_t = \beta_1 m_t + (1 - \beta_1)\Delta_t$
$\quad v_t = \beta_2 v_t + (1 - \beta_2)\Delta_t^2$
$\quad x_t = x_t + \eta \frac{m_t}{\sqrt{v_t} + \tau}$
**end**

---

## H  MOMENTUM-BASED SERVER-SIDE OPTIMIZERS AND BATCH NORMALIZATION

Certain layers in deep learning models accumulate state throughout training that must remain strictly positive. Batch normalization (BN) (Ioffe & Szegedy, 2015) is one of these layers. During training, the layer accumulates mean and variance estimates of preceding layer activations. These estimates are used to perform normalization during inference. When using FedAvg, these estimates are aggregated, like all other parameters, using linearly-weighted averaging. Thus, strictly positive local quantities result in strictly positive aggregated quantities. However, when using momentum-based server-side optimization, as in FedAdam, this property is no longer preserved. This is particularly problematic for BN, where the square root of the variance is computed to perform normalization.

As an illustration of this phenomenon, consider learning a single parameter $x$ and an initial guess of $x_0 = 2.0$. Set the FedAdam parameters to $\beta_1 = 0.9$, $\beta_2 = 0.9$, $\tau = 1\text{e-}9$, and a learning rate of $\eta = 0.1$. Assume that each client calculates a new parameter of $\tilde{x} = 0.1$ at each iteration. Algorithm 1 shows the iterative calculations of FedAdam for this simplified setting. Running these iterations results in a negative value for $x_t = -0.204$, despite $\tilde{x} > 0$ for every iteration. This drift is due to the momentum associated with the updates. Therefore, in settings where BN is present and the

FedAdam strategy is employed, batch tracking is shut off to avoid such issues. It should be noted that this is qualitatively similar to the FedBN strategy in (Li et al., 2021).

## I    EXTREME HETEROGENEITY STUDY

As a test of the robustness of the FL methods studied here, with particular interest in FENDA-FL, an experiment inducing extreme feature-space heterogeneity is designed. For the GEMINI Delirium task, described in Section 3.2, each client further processes its data with dimensionality reduction through PCA and feature selection to produce input vectors of dimension 300. As the clients independently perform this process, the resulting feature spaces are no longer well-aligned. In this setup, locally trained models are unlikely to generalize well to other domains. Moreover, FL techniques that produce a single shared model are expected to encounter convergence issues. This is confirmed in the results shown in Figure 5. Each client's local model generalizes poorly to other domains and global FL approaches struggle with performance. On the other hand, the personalized FL approaches fare better, as they are able to lean on local training when global parameters are not informative. FENDA-FL demonstrates notable robustness to this setting as the only approach to recover the siloed performance target. Note also that local checkpointing is important for this task.

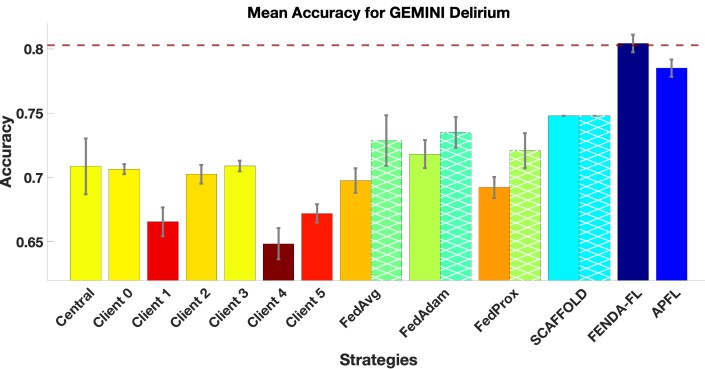

Figure 5: Mean accuracy across clients for the GEMINI Delirium task when each client performs independent feature processing. The dashed horizontal line represents siloed performance, see Section 3.3. Solid and hatched bars indicate global and local checkpointing, respectively.

## J    FENDA-FL TECHNICAL DETAILS AND DISTRIBUTION SHIFT

Assume there is a central server and $M$ clients, each with a local dataset $D_i$, for $i \in \{1, \ldots, M\}$. Client data exists on domain $X \times Y$ where $X \in \mathbb{R}^I$ and $Y \in \mathbb{R}^O$. FENDA-FL uses the FedAvg algorithm as a foundation to enable collaborative learning among clients. It extends FedAvg by taking a personalized approach inspired by FENDA. In particular, along with a global feature extractor $f_{\mathbf{w}} : X \to \mathbb{R}^{H_G}$, each client, $i$, has a local feature extractor $f_{\mathbf{w}_i} : X \to \mathbb{R}^{H_i}$ and prediction head $g_i : \mathbb{R}^{(H_G+H_i)} \to Y$. For a given input sample $\mathbf{x} \sim P_i(X)$, $f$ and $f_i$ map $\mathbf{x}$ to representations $\mathbf{z}$ and $\mathbf{z}_i$, that are concatenated and input into $g_i$ to yield a prediction, $\hat{y}_i$. In the simplest case, $g_i$ is a fully connected layer parameterized by weights $\theta_i \in \mathbb{R}^{O \times (H_G+H_i)}$ and an activation function $\sigma$ such that

$$\hat{y}_i = \sigma \left( \theta_i \begin{bmatrix} \mathbf{z} \\ \mathbf{z}_i \end{bmatrix} \right) = \sigma \left( \theta_i \begin{bmatrix} f_{\mathbf{w}}(\mathbf{x}) \\ f_{\mathbf{w}_i}(\mathbf{x}) \end{bmatrix} \right).$$

The global feature extractor weights, $\mathbf{w}$, are aggregated across clients via FedAvg, whereas the local feature extractor and prediction head weights, $\mathbf{w}_i$ and $\theta_i$, are exclusively learned through SGD, or its variants, on each client. Both global and local parameters are iteratively updated across FL rounds.

### J.1    DISTRIBUTION SHIFTS IN FL

Distribution shifts arise when the joint distribution of inputs and targets varies between datasets (Quinonero-Candela et al., 2008). In the standard ML setup, distribution shifts occur between the

train and test data, such that $P_{\text{train}}(X, Y) \neq P_{\text{test}}(X, Y)$, leading to poor model generalization. The most common distribution shifts include label, covariate and concept drift (Zhang et al., 2021). Label shift is characterized by different label distributions $P(Y)$, but constant class conditional distribution, $P(X|Y)$. Covariate shift occurs when the marginal distribution, $P(X)$, varies across datasets while the conditional distribution, $P(Y|X)$, remains the same. Finally, concept drift arises when the class conditional distribution $P(Y|X)$ varies and $P(Y)$ remains fixed.

In the FL setting, distribution shifts are common among clients, due to variations in the underlying data generating process. These variations may stem, for example, from differences between the sensors, preferences, geographies and time periods involved in data collection for each client (Kairouz et al., 2021). Such shifts are often referred to as heterogeneous or non-IID data. In this regime, FedAvg often exhibits slow convergence and poor performance. Many works have attempted to address this for various types of shifts (Karimireddy et al., 2020; Deng et al., 2020; Li et al., 2021).

As a personalized method, FENDA-FL allows each client to train a custom model while maintaining the benefit of using FL to jointly learn a subset of weights. In particular, as described above, FENDA-FL has a locally-optimized feature extractor and prediction head, in addition to a global feature extractor learned via FedAvg. The local feature extractor and prediction head endow client models with a degree of resilience to label and covariate shift. They may adapt feature representations and predictions to lean more or less heavily on global representations or modify prediction distributions to match local statistics. Beyond such shifts, FENDA-FL is able to handle concept drift where the conditional distribution, $P(Y|X)$, varies across clients. For example, in the extreme case, this is accomplished by the local prediction head ignoring global features altogether. This is not true of non-personalized FL models, which assume a single function exists that obtains low error across all clients.

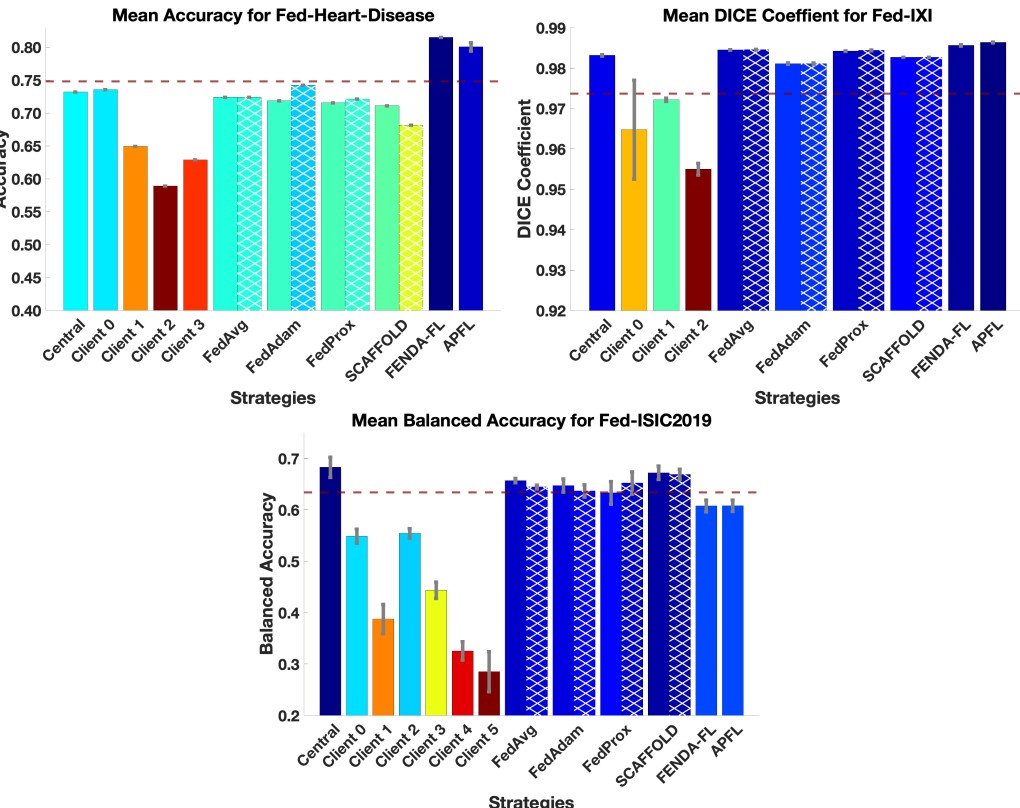

Figure 6: Each figure shows the average performance for various training techniques on the FLamby tasks. The dashed horizontal line represents siloed performance, as described in Section 3.3. Error bars illustrate 95% confidence intervals. Solid and hatched bars indicate global and local checkpointing, respectively.

FENDA-FL is also resilient to extreme distribution shifts involving misalignment of client features. As demonstrated in Appendix I, FENDA-FL is able to obtain performance consistent with locally trained models when the feature spaces of the clients are not aligned, where as other benchmark approaches fall significantly short. One type of distribution shift that is not admissible in the proposed approach is misalign labels spaces. This is an important setting to explore in future work.

## K VISUALIZATIONS OF FLAMBY AND GEMINI RESULTS

In this section, we present an additional view of the performance of the strategies studied in Section 4. Figure 6 present the results of Table 2 and Figure 7 displays those of Tables 3, and 4.

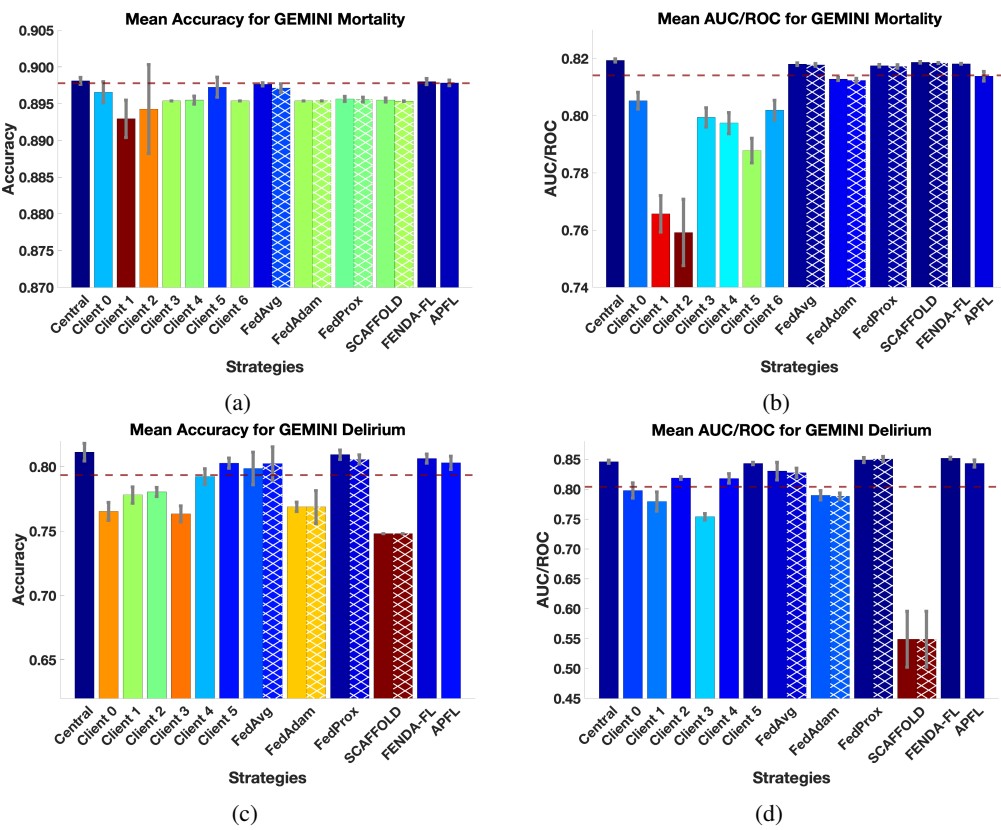

Figure 7: Each figure shows the average performance for various training techniques on the GEMINI tasks. The dashed horizontal line represents siloed performance, as described in Section 3.3. Error bars illustrate 95% confidence intervals. Solid and hatched bars indicate global and local checkpointing, respectively.

