# OpenReview forum: "FENDA-FL: Personalized Federated Learning on Heterogeneous Clinical Datasets"
_ICLR.cc/2024/Conference — Submitted to ICLR 2024_

### Official Review · Reviewer_ewfD · 2023-10-28

**Soundness:** 1 poor
**Presentation:** 3 good
**Contribution:** 1 poor
**Rating:** 3
**Confidence:** 4

**Summary:**

In this study, the authors introduce a federated domain adaptation model designed to train personalized FL models. The model in question is an adaptation of the FENDA domain adaptation method, leveraging both a global feature extractor and a local feature extractor to adapt data across different domains. Evaluations of the model have been conducted on an FL benchmark dataset as well as a real-world clinical dataset.

**Strengths:**

The experimental design is thorough, and the paper is well-written and easy to follow.

**Weaknesses:**

My major concern is a significant oversight in this work, which is the absence of discussions of an important line of related works - federated domain adaptation/generalization. This topic is closely related to the central topic of this paper. There are many advanced techniques for federated domain adaptation/generalization in recent years, for examples [1-2] and numerous other contributions in this domain. Yet, the authors seem to have omitted any discussion contrasting their proposed model with these works, nor have they incorporated them as baseline models for comparison. The decision to utilize the FENDA model, which appears potentially outdated in the domain adaptation field, raises concerns. Without comparing the proposed model against current SOTA methods in the field of federated domain adaptation, it is hard to assert that the domain adaptation strategy showcased here represents the state-of-the-art in FL.

[1] Yao, C. H., Gong, B., Qi, H., Cui, Y., Zhu, Y., & Yang, M. H. (2022). Federated multi-target domain adaptation. In Proceedings of the IEEE/CVF Winter Conference on Applications of Computer Vision (pp. 1424-1433).
[2] Zhang, R., Xu, Q., Yao, J., Zhang, Y., Tian, Q., & Wang, Y. (2023). Federated domain generalization with generalization adjustment. In Proceedings of the IEEE/CVF Conference on Computer Vision and Pattern Recognition (pp. 3954-3963).

**Questions:**

Please address the weaknesses above.

---

> ### Author Response · Authors · 2023-11-20
>
> Thank you for taking the time to review our work and providing useful feedback on how our paper might be improved. We have significantly revised the paper to incorporate new experiments and discussion. Below we discuss each of the comments and outline how the revision aims to address them.
>
> _My major concern is a significant oversight in this work, which is the absence of discussions of an important line of related works - federated domain adaptation/generalization. This topic is closely related to the central topic of this paper. There are many advanced techniques for federated domain adaptation/generalization in recent years, for examples [1-2] and numerous other contributions in this domain. Yet, the authors seem to have omitted any discussion contrasting their proposed model with these works, nor have they incorporated them as baseline models for comparison. The decision to utilize the FENDA model, which appears potentially outdated in the domain adaptation field, raises concerns. Without comparing the proposed model against current SOTA methods in the field of federated domain adaptation, it is hard to assert that the domain adaptation strategy showcased here represents the state-of-the-art in FL._
>
> _[1] Yao, C. H., Gong, B., Qi, H., Cui, Y., Zhu, Y., & Yang, M. H. (2022). Federated multi-target domain adaptation. In Proceedings of the IEEE/CVF Winter Conference on Applications of Computer Vision (pp. 1424-1433). [2] Zhang, R., Xu, Q., Yao, J., Zhang, Y., Tian, Q., & Wang, Y. (2023). Federated domain generalization with generalization adjustment. In Proceedings of the IEEE/CVF Conference on Computer Vision and Pattern Recognition (pp. 3954-3963)._
>
> In the revision, we have included a thorough discussion of the intersection and differences between federated domain adaptation and personalized FL, which we consider here. The majority of this discussion occurs in Section 2 and Appendix J. As noted, personalized FL and federated domain adaptation are closely related fields. However, the settings that are considered differ in some significant ways. For example, [1] considers a server-hosted, labeled source dataset and client-based, unlabelled target datasets, which is quite distinct from the setting considered in our work. As such, [1] does not include benchmarks against standard FL approaches. Similarly, other works in federated domain adaptation, such as [3, 4, 5], also consider distinct settings with a mixture labeled and unlabeled datasets, distributed in various ways.
>
> In this paper, we do not explicitly consider domain adaptation as the target task. Rather, we apply a technique from domain adaptation as a means of improving the robustness of personalized FL in the context of non-IID or heterogeneous datasets. However, the Generalization Adjustment approach proposed in [2] is very interesting. As an augmentation to existing FL approaches, it appears quite promising. The primary evaluation criterion in [2] is “leave-one-domain-out,” where only a subset of clients are used in training and generalization is measured exclusively on the held-out client domain, which is somewhat distinct from our setting. However, using it to improve both global and personal FL approaches is intriguing. We have included it as an important avenue for future work.
>
> [1] Yao, C. H., Gong, B., Qi, H., Cui, Y., Zhu, Y., & Yang, M. H. (2022). Federated multi-target domain adaptation. In Proceedings of the IEEE/CVF Winter Conference on Applications of Computer Vision (pp. 1424-1433).
>
> [2] Zhang, R., Xu, Q., Yao, J., Zhang, Y., Tian, Q., & Wang, Y. (2023). Federated domain generalization with generalization adjustment. In Proceedings of the IEEE/CVF Conference on Computer Vision and Pattern Recognition (pp. 3954-3963).
>
> [3] X. Peng, Z. Huang, Y. Zhu, and K. Saenko. Federated adversarial domain adaptation. arXiv preprint arXiv:1911.02054, 2019.
>
> [4] L. Song, C. Ma, G. Zhang, and Y. Zhang. Privacy-preserving unsupervised domain adaptation in federated setting. IEEE Access, 8:143233–143240, 2020. Doi: 10.1109/ACCESS.2020.3014264.
>
> [5] Y. Shen, J. Du, H. Zhao, Z. Ji, C. Ma, and M. Gao. FedMM: A communication efficient solver for federated adversarial domain adaptation. In Proceedings of the 2023 International Conference on Autonomous Agents and Multiagent Systems, AAMAS ’23, pp. 1808–1816, Richland, SC, 2023. International Foundation for Autonomous Agents and Multiagent Systems. ISBN 9781450394321

---

> ### Comment · Reviewer_ewfD · 2023-11-22
>
> Thanks to the authors for their response. However, I'll stand for my original score due to the limited related work discussion and baseline comparison.

---

### Official Review · Reviewer_Hvzv · 2023-10-30

**Soundness:** 2 fair
**Presentation:** 3 good
**Contribution:** 2 fair
**Rating:** 3
**Confidence:** 5

**Summary:**

This paper adapts the FENDA method to the federated setting. The central concept involves maintaining both local feature extractors and global feature extractors for each client, with the central server utilizing Fedavg to aggregate the weights of the global feature extractors from each client. The paper rigorously assesses the proposed method using benchmarks and real-world tasks, enhancing evaluation robustness by conscientiously selecting checkpoints based on cross-validation and employing additional baseline comparisons.

**Strengths:**

- The method is comprehensible and straightforward.
- The experiments meticulously evaluate the proposed method by selecting the appropriate checkpoints, which are convincing and robust. The real-world clinical scenarios are well-suited for federated learning.

**Weaknesses:**

- This paper addresses the challenge of personalized federated learning using heterogeneous local datasets; however, the mechanism is not thoroughly explained. The proposed method draws inspiration from domain adaptation, where distribution shifts naturally occur. In this paper, local clients have datasets sampled from various distributions. The question arises: What types of distribution shifts (such as feature shifts or label shifts) can the proposed method effectively handle?

**Questions:**

Please describe the mechanism of the proposed method in detail and elucidate its effectiveness in addressing different types of distribution shifts, particularly in the context of client dataset heterogeneity.

---

> ### Author Response · Authors · 2023-11-20
>
> Thank you for your attentive review of our work. We appreciate your time and the opportunity to improve our results along the lines of your suggestions. We have endeavored to revise the paper to thoroughly address the weaknesses identified in the review. Below we discuss each of the comments and outline how the revision aims to address them.
>
> _The main text of the submission exceeds 9 pages._
>
> In the submission guidelines, authors were encouraged to include a “Reproducibility Statement” at the end of the main text but before the “References.” If we have read the instructions properly, the statement is not counted towards the page limit. Since it is the “Reproducibility” section that exceeds 9 pages, we believe that the submission is still compliant. However, if we are incorrect in such an interpretation, we would appreciate the correction and will ensure that the revision is, in fact, compliant.
>
> _This paper addresses the challenge of personalized federated learning using heterogeneous local datasets; however, the mechanism is not thoroughly explained. The proposed method draws inspiration from domain adaptation, where distribution shifts naturally occur. In this paper, local clients have datasets sampled from various distributions. The question arises: What types of distribution shifts (such as feature shifts or label shifts) can the proposed method effectively handle?_
>
> _Please describe the mechanism of the proposed method in detail and elucidate its effectiveness in addressing different types of distribution shifts, particularly in the context of client dataset heterogeneity._
>
> We agree that such a discussion is important and was not adequately present in our original draft. A formal description of the FENDA-FL approach is now included in Appendix J. Therein, we also discuss the admissibility of the three most common types of distribution changes: label shift, covariate shift, and concept drift. Because FENDA-FL includes a locally trained feature extractor and classification head, each of these shifts can theoretically be handled by the approach. For label and covariate shift, these modules may adapt feature representations and predictions to lean more or less heavily on global representations or modify prediction distributions to match local statistics. For concept drift, the relationship between labels and global features may be de-emphasized in favor of local representations. In extreme cases, where global information is not relevant, the local prediction head may learn to ignore global features altogether. We hope that this section more thoroughly, and clearly, describes the FENDA-FL design and its flexibility in non-IID data settings.

---

> > ### Comment · Reviewer_Hvzv · 2023-11-23
> >
> > Thank you for the author's reply. The discussion on handling different types of distribution shifts is clearer now, but it still needs further elaboration. It would be beneficial to include some simple toy experiments to illustrate these mechanism. I will keep my score unchanged, as the method has the potential to be more novel.

---

### Official Review · Reviewer_ubEK · 2023-10-31

**Soundness:** 3 good
**Presentation:** 3 good
**Contribution:** 3 good
**Rating:** 5
**Confidence:** 3

**Summary:**

This work expands on the federated learning space in clinincal domain by extending the FENDA method (a domain adaptation method) to federated learning, while showing improvements on the FLamby benchmark.

**Strengths:**

- Clear presentation, well written paper.
- Appropriate comparison of baselines on the FLamby benchmark.
- Extensive analysis of multiple other baselines

**Weaknesses:**

- It would be nicer to also see a table of the results, rather than a visual representation due to the number of settings. Minor comment: The color choices for the bar graphs seem to represent goodness, with red being worse and blue being better; however, the different colors are a little jarring and could cause confusion.
- The performance benefit compared to APFL shown in Fig. 2 is not very clear, also in Fig. 3, a similar obersvation can be made for FedAVG and APFL

**Questions:**

- The idea of using domain adaptation seems like a natural pairing with FL. Can the authors include a discussion on the inherent differences / relationships between these 2 fields for better context? Google scholar reveals some papers such as https://arxiv.org/abs/1911.02054 and https://arxiv.org/abs/1912.06733 which may be relevant.
- Is it possible to release a version of the code that works for open-source datasets?

---

> ### Author Response · Authors · 2023-11-20
>
> Thank you for your review of our work and offering your thoughts on where the paper could be improved. We have attempted to address each of the concerns raised and believe that the paper is markedly improved as a result. Below we discuss each of the comments and outline how the revision aims to address them.
>
> _It would be nicer to also see a table of the results, rather than a visual representation due to the number of settings. Minor comment: The color choices for the bar graphs seem to represent goodness, with red being worse and blue being better; however, the different colors are a little jarring and could cause confusion._
>
> Thank you for your suggestion. Other reviewers also noted that the bar graphs did not present our results as clearly as we had hoped. Taking your suggestion, we have modified the presentation of the results to be in a tabular format. We hope that the representation of the results is improved. The bar graphs are still presented in Appendix K. However, if they should be removed, we are happy to do so.
>
> _The performance benefit compared to APFL shown in Fig. 2 is not very clear, also in Fig. 3, a similar observation can be made for FedAVG and APFL_
>
> With the change of results presentation, we hope that the readability and ease of analysis has been measurably improved. The experimental results are now presented in Tables 2-4. FENDA-FL outperforms APFL on the Fed-Heart-Disease task by a significant margin. While FENDA-FL is the second best method for Fed-IXI, the new architecture studies in Appendix F show that the gap between the two methods can be closed through thoughtful network design. For Fed-ISIC2019, each method under-performs global approaches. For both GEMINI Mortality and Delirium, FENDA-FL outperforms APFL in all metrics.
>
> _The idea of using domain adaptation seems like a natural pairing with FL. Can the authors include a discussion on the inherent differences / relationships between these 2 fields for better context? Google scholar reveals some papers such as https://arxiv.org/abs/1911.02054 and https://arxiv.org/abs/1912.06733 which may be relevant._
>
> We agree that the pairing of techniques in domain adaptation and federated learning is quite natural. In the design of FENDA-FL, we have borrowed an approach from domain adaptation to improve personalized FL and aim to continue to do so in future work. Generally, domain adaptation (DA), including federated domain adaptation, considers the setting of training a model using labeled source datasets and unlabeled, or sparsely labeled, target datasets such that the model performs well in the target domain. As such, the space considers a distinct setting from that considered here but deals with similar data drift issues. For example, in [1], the server holds an unlabeled target dataset while labeled client-hosted datasets serve as the source.
>
> In the revision, we have highlighted the differences and intersections between the disciplines in Section 2, including a discussion of the references provided and those suggested by Reviewer #4. Interestingly, [2] uses a slight variation of APFL and applies DP-SGD for privacy during client training and, as such, does not consider a true DA setting. We have included a reference to this work under settings not considered because the main difference is the use of DP-SGD in client-side training.
>
> [1] X. Peng, Z. Huang, Y. Zhu, and K. Saenko. Federated adversarial domain adaptation. arXiv preprint arXiv:1911.02054, 2019.
>
> [2] D. W. Peterson, P. Kanani, and V. J. Marathe. Private federated learning with domain adaptation. In the 33rd Conference on Neural Information Processing Systems (NeurIPS 2019), Proceedings of NeurIPS, Vancouver, Canada, 2019.
>
> _Is it possible to release a version of the code that works for open-source datasets?_
>
> This is very much possible and our intention! In releasing the experimental code associated with the paper, which works with the open-source datasets in FLamby, we will also be making the underlying library, which is not specific to the datasets studied, publicly available. The library is compatible with any open-source dataset that can be used with pytorch. Moreover, the library includes a number of examples to facilitate using the implemented methods for other experiments. Our hope is that this library will accelerate research in FL by making experimentation easier and the implementation of new techniques more straightforward.

---

> > ### Comment · Reviewer_ubEK · 2023-11-22
> >
> > Thank you for clarifying those tables and the ablations. I am planning on maintaining my score as a whole since I believe it is still a fair evaluation of the paper in general. However, I do have some followup comments.
> >
> > - For Table 2: It was confusing how some clients do not have local scores for Fed-Heart-Disease and Fed-IXI, but do have them for Fed-ISIC2019. Please state that this is explicitly because of the true number of clients in the dataset in the Table caption and mark the cells with "-" for placeholder.
> >
> > - For Table 3, APFL and FENDA-FL are still within the 95\% confidence interval for accuracy. I believe the claim that FENDA-FL outperforms APFL in all metrics could and should be further investigated.

---

> > > ### Author Response · Authors · 2023-11-22
> > >
> > > Thank you for the additional feedback on the presentation of Table 2 and the analysis of Table 3. We have updated Table 2 along the line you suggested. For Table 3, we have softened the claim of FENDA-FL outperforming APFL in all metrics to include acknowledgement that the improved accuracy is not necessarily statistically significant. These changes are reflected in our updated draft.

---

### Official Review · Reviewer_kWTA · 2023-11-01

**Soundness:** 2 fair
**Presentation:** 2 fair
**Contribution:** 2 fair
**Rating:** 3
**Confidence:** 4

**Summary:**

The authors introduce FENDA-FL, an adaptation of the Frustratingly Easy Neural Domain Adaptation (FENDA) method for FL, focusing on the personalized FL paradigm where each participant trains a model tailored to their local data distribution. The effectiveness of FENDA-FL is demonstrated through comprehensive experiments using various clinically relevant datasets, including a subset from the FLamby benchmark and tasks from the GEMINI datasets.

**Strengths:**

- The application of the FENDA method, originally used for domain adaptation, to Federated Learning (FL) is effective, aligning domain-agnostic features with FL's global components and domain-specific traits with local elements.

**Weaknesses:**

- While adapting FENDA's concept to the FL scenario required some modifications, the application of FENDA in this methodology appears overly incremental. The approach of combining features to utilize both global and local information, as opposed to APFL's method of aggregating logits, is not particularly novel.
- The methodological contribution to the ICLR community unclear. The application of FENDA to federated learning is straightforward, "federated checkpointing" is conceptually a combination early stopping with federated learning.
- The paper lacks a thorough ablation study on federated checkpointing methods. Additionally, while it benefits from not restricting the network architectures between global and local feature extractors, it would have been informative to show performance variations with the use of diverse network architectures.
- The paper could greatly improve the visibility of performance differences between methodologies in its figures. Currently, the color-coding for each method is not distinct enough, and the representation of all performances via bar graphs makes it challenging to discern if the differences, including standard deviations, are statistically significant. In terms of visibility, it falls significantly short.

**Questions:**

Please see the weaknesses

---

> ### Author Response · Authors · 2023-11-20
>
> Thank you for taking the time to carefully read and review our manuscript and for the thoughtful suggestions on how the work could be improved. We have conducted several new experiments along the lines suggested and worked to significantly improve the presentation of our results. Below we discuss each of the comments in order and outline how the revision aims to address them.
>
> _While adapting FENDA's concept to the FL scenario required some modifications, the application of FENDA in this methodology appears overly incremental. The approach of combining features to utilize both global and local information, as opposed to APFL's method of aggregating logits, is not particularly novel._
>
> While the architectural differences between APFL and the FENDA-FL approach are not large, there are several significant benefits to the FENDA-FL design that we believe make the proposed modifications important. First and foremost, FENDA-FL outperforms APFL in many of the experiments and is quite competitive in the cases that it does not. Conceptually, the move from the aggregation of logits to the synthesis of feature representations is fairly straightforward. However, the change also implies that training can be done monolithically and that global and local module combinations can incorporate non-linear transformations rather than just weighted averaging.
>
> As noted below, the approach also provides significant flexibility in the design of the local and global feature extraction modules and does not require additional hyper-parameters. In APFL, the local and global modules are exact copies, which limits any inductive bias one might inject into the module designs. As pointed out in another comment, we did not demonstrate the advantage of this flexibility as well as we could have. Appendix F now contains results for studies considering the benefit of adding inductive bias to the FENDA-FL model design.
>
> Finally, the move to the FENDA-FL design is well founded on the success of FENDA. In our revision, we have endeavored to make these improvements more clear, while also recognizing that the change in design is straightforward.
>
> _The methodological contribution to the ICLR community unclear. The application of FENDA to federated learning is straightforward, "federated checkpointing" is conceptually a combination early stopping with federated learning._
>
> Thank you for the opportunity to emphasize the contributions of this work. While the adaptation of FENDA is straightforward, we believe it is a useful conceptual bridge between personalized FL methods and domain adaptation. In future work, we hope to bring additional ideas from domain adaptation into the setting of personalized FL. With respect to checkpointing, the proposed characterization is certainly applicable.  However, early stopping considers terminating training early, while in the proposed approach federated training continues, with other clients potentially benefiting from further training. Most existing works simply perform a fixed number of server rounds before evaluation.
>
> While the approach is fairly simple, it helps mitigate the risk of overfitting, especially when client data imbalance is an issue. The combination of federated checkpointing and our proposed evaluation criterion aim to bring FL evaluation more in line with standard ML practices. As suggested in another comment, we have conducted experiments without federated checkpointing to demonstrate the advantage of its use. These results are included in Appendix E in the revision.
>
> When considered as a whole, we believe that the experimental framework and results presented here constitute an important representation of effective and fair benchmarking in FL studies. We offer several improvements to evaluation, including checkpointing and various lenses through which FL methods should be evaluated. Finally, the experiments on the FLamby benchmark establish significant improvements for many methods on the selected tasks.
>
> (Discussion Continued in a follow up comment)

---

> > ### Author Response · Authors · 2023-11-20
> >
> > _The paper lacks a thorough ablation study on federated checkpointing methods. Additionally, while it benefits from not restricting the network architectures between global and local feature extractors, it would have been informative to show performance variations with the use of diverse network architectures._
> >
> > These are extremely valuable suggestions. Ablation studies of federated checkpointing for Fed-Heart-Disease and Fed-IXI are now reported in Appendix E, demonstrating the benefit of the method in several scenarios. Further, experiments studying the effects of architecture variations for the global and local feature extractors in FENDA-FL are discussed in Appendix F. These variations include modifying the size of the underlying feature extractors and the corresponding latent spaces. The experiments demonstrate the benefits that can be seen when injecting inductive bias into the FENDA-FL architecture design.
> >
> > _The paper could greatly improve the visibility of performance differences between methodologies in its figures. Currently, the color-coding for each method is not distinct enough, and the representation of all performances via bar graphs makes it challenging to discern if the differences, including standard deviations, are statistically significant. In terms of visibility, it falls significantly short._
> >
> > Thank you for sharing this with us. This was also noted by other reviewers, and we have taken the opportunity to redesign the reporting of the results as tables. We hope this has markedly improved the readability and interpretability of our results.

---

### Meta-Review · Area_Chair_Ko9V · 2023-11-30

**Metareview:**

In the context of heterogeneous local data in federated learning, the paper proposes a personalized approach based on domain adaptation, based on a work of Kim et al in 2016 (FENDA). The method is evaluated on benchmarks in the medical domain.

The paper is well written and benefits from the FLamby and the Gemini benchmarks. The main weakness is the lack of novelty since the FENDA extension is rather direct. Another point is the positioning of the domain adaptation approach in personalized FL. The authors have added some material to better explain the algorithm itself and this relationship, but some additional work is needed to reach the acceptance level of ICLR.

**Justification For Why Not Higher Score:**

There is agreement on rejection in all the reviews.

**Justification For Why Not Lower Score:**

N/A

---

### Decision · Program_Chairs · 2024-01-16

Reject